# Social validity of acceptance-based workplace mental health training for use in a low resource setting. *A qualitative study with Ugandan mental health providers*

**Khamisi Musanje**[1]\*, **Paul E. Flaxman**[2], **Ross McIntosh**[2], **Rosco Kasujja**[1]

**1** School of Psychology, Makerere University, Kampala, Uganda, **2** School of Psychology, City University of London, London, United Kingdom

\* Khamisi.musanje@mak.ac.ug

**Data Availability Statement:** The data supporting the conclusions of the study is attached as supplementary material to the manuscript.

## Abstract

In low-resource settings, working age adults suffer a disproportionately higher mental health burden due to precarious work environments; yet access to evidence-based worksite mental health promotion programs remains severely limited due to the lack of professional service providers. Alternative mental wellness programs that are affordable, accessible and help build resilience to deal with the daily demands of work and life need to be introduced in workplaces of low-resource contexts. Such programs should however be acceptable and of social importance to the targeted contexts. Acceptance-based approaches meet these criteria but have mainly been implemented and evaluated in high-income countries. Gauging the appropriateness of such programs for low-resource workplace settings before wide-scale implementation is necessary. We engaged n = 14 mental health providers living and working in Kampala-Uganda in a one-day workshop focused on using acceptance and commitment training in the workplace. During in-depth interviews, these providers shared feedback on the social importance of the program's goals and effects, and acceptability of the program's procedures to Ugandan society. A deductive thematic approach was used to analyze data (codes organized according to *a priori* thematic categories that aligned with the Social Validity Framework). Findings showed that the program's goals and effects were gauged as socially significant and the training process was acceptable. However, key adjustments were recommended, including introducing communal values into the program, reducing session load, integrating mental health awareness strategies, and adding contextually relevant metaphors. These adjustments are likely to enhance the appropriateness of this type of acceptance-based worksite mental health promotion program for use in Uganda.

## Introduction

Globally, an estimated 15% of all working adults suffer mental health-related challenges, costing approximately US$ 1 trillion and 12 billion working days annually [1]. The mental health burden appears even higher in low and middle-income countries (LAMICs), where 75% of the world's labour force lives [2]. The work environment in LAMICs has been classified as

**Funding:** This research was supported by Research England's Higher Education Innovation Fund (HEIF), via City, University of London (CUL). https://www.ukri.org. PF received the support. "The funders had no role in study design, data collection and analysis, decision to publish, or preparation of the manuscript."

precarious [3], characterized by non-regulated work arrangements, exploitation, poor working conditions, long working hours, and work overload [3]. The situation is further exacerbated by high rates of unemployment, which make the negotiation of welfare difficult [4]. Yet, persons of working age in LAMICs spend a significant proportion of their lifetime at work [1], potentially compromising their mental health and quality of life [5].

WHO recommends the provision of appropriate and affordable workplace-based mental health support services to workers in LAMICs as part of employee assistance programs to de-escalate the mental health burden [6,7]. However, evidence of implementation and utilization of such support interventions in LAMICs is limited [7]. In Uganda, while 68% of the working population is estimated to be experiencing common mental health challenges [8], over 78% of employers have never offered any form of mental health support to employees [9]. The shortage of resources to support mental health initiatives, the volatile operating environment, competing organizational priorities, limited number of mental health service providers, and lack of standardized evidence-based psychological interventions have been highlighted as key challenges to integrating mental health support services into workplaces in LAMICs [8,10].

Improving working conditions would likely lessen the mental health burden at work [11]; however, the scarcity of resources in LAMICs makes such structural adjustment difficult, at least in the short term [12]. Thus, implementing innovative and scalable mental health promotion programs, which empower workers to deal with the demands of a precarious work environment, may offer alternative solutions to the growing mental health burden [10]. To be scalable, such interventions ideally need to be transdiagnostic, so that they can simultaneously address the range of common mental health challenges found in the LAMICs' workforce [13]. Interventions also need to be cost-effective (e.g., by promoting self-management skills), given the limited number of mental health professionals [4]. Finally, worksite programs should help workers develop personal resilience resources that have reliably been shown to offer some protection against the mental health impact of work-related stressors [7].

In recent years, there has been interest in the potential use of interventions based on *acceptance and commitment therapy* (ACT) for tackling common mental health problems among specific communities in LMICs ([13–16]. ACT is part of the broader field of contextual behavioral science (CBS), and has some distinctive features that may facilitate adoption in low-resource contexts [16]. In support of ACT's suitability for such contexts, a number of feasibility or pilot efficacy trials conducted in various LMICs have indicated that ACT-based interventions can improve markers of mental health and well-being [14]. In high income countries, the workplace has long been utilized as an important arena for delivering ACT-based programs, aimed at addressing (or preventing) low intensity mental health difficulties and improving psychological well-being in the general population ([17–22]. However, to date, the potential of disseminating ACT skills and principles via workplaces in LMICs appears to have been largely overlooked. This is likely due to the lack of culturally sensitive translations of the ACT-based training protocols that have thus far proved viable and effective primarily in Western nations. Accordingly, the aims of the current study are to 1) scrutinize the social validity of a workplace version of ACT for potential use in Uganda, and 2) gather local stakeholders' perspectives on specific ways in which an existing workplace ACT protocol might need to be tailored to ensure relevance among the Ugandan workforce.

## Literature review

There is an emergent and growing scholarly literature highlighting the potential of deploying ACT-based interventions as part of global mental health initiatives [14]. Regardless of context, ACT interventions seek to improve people's ability to be psychologically flexible, which is

broadly defined as a willingness to experience difficult or discomforting thoughts and feelings while pursing personally valued patterns of behavior [23]. ACT has some features that are proposed to support cross-cultural applicability, including a transdiagnostic philosophy, feasibility for delivery via guided self-help methods and by non-specialist healthcare facilitators, and emphasis on the development of psychological skills, psychological flexibility, and values-based living, rather than the elimination of difficult internal states [14,16]. These features of the ACT approach have been recognized by the World Health Organization (WHO), which supported the development of self-help versions of ACT for use among LMIC communities facing adversity [13]. A small body of research has supported the feasibility and effectiveness of ACT-based initiatives in low resource settings. For example, among the African nations, the aforementioned WHO self-help ACT-based program was shown to reduce psychological distress and improve subjective well-being among South Sudanese women in refugee settlements in Uganda [15]. In Sierra Leone, an ACT-based training course was found to increase psychological flexibility and life satisfaction among non-specialist healthcare workers [24]. In Nigeria, ACT interventions have exhibited viability for improving mental health and psychosocial functioning among HIV+ pregnant women [25], spiritually abused adolescents [26], and individuals experiencing substance misuse problems [27].

The rationale underpinning these initiatives is that ACT is a scalable approach, which could be widely disseminated as a low-intensity psychological intervention for reducing or preventing common mental health problems (such as anxiety and depression) and cultivate an improved sense of personal, emotional, and social well-being [14,16]. It is noteworthy that this same rationale underpins ACT's widespread application in workplace settings in many of the world's high-income countries, in which the aim is to reduce psychological distress and enhance well-being among members of the general working population [19]. A set of recent reviews highlighted broadly positive effects of ACT's workplace applications. For example, in a meta-analytic review, ACT was found to be effective in improving healthcare professionals' general and work-related mental health over and above control conditions [20]. Likewise, ACT was effective in improving mental health and well-being among office-based workers [22]. Finally, delivering ACT to employees often reduces the risk of job burnout [21]. Aside from the lack of culturally adapted workplace ACT training protocols, there appear to be no significant barriers to extending these benefits to workers located in LMICs. The current study represents an initial step in this endeavor, by assessing the viability of adapting and delivering a workplace ACT program for employees in Uganda.

## The present study: Assessing the social validity of an ACT-based training protocol for Ugandan workplaces

As already noted, ACT-based training programs are increasingly implemented and evaluated in workplace settings in high-income countries (e.g., the UK, Australia, Canada, US, Spain, and the Nordic countries), particularly (although not exclusively) for healthcare staff [20,28–30]. ACT-W seeks to improve workers' mental health by supporting the development of psychological flexibility. [31] Evidence indicates that psychologically flexible workers tend to report better mental health, good quality of life, and are more productive [32,33].

Accordingly, using ACT-W in a low-resource setting, such as Uganda, might help to protect and improve the mental health of workers and consequently result in a better quality of life. However, most interventions developed for use in high-income contexts, which are characterized by clear work arrangements, better working conditions, highly educated workers, and regulated employment relationships, may be insufficiently tailored for delivery in low-resource settings [34]. Thus, the feasibility, and cultural relevance of goals, processes, and anticipated

effects of ACT-W need to be validated in the Ugandan context to ascertain fit or misfit prior to wider implementation [35].

Assessing the appropriateness of intervention goals, effects on the needs of a society, and acceptability of its procedures to the targeted population, is conceptualized as establishing *social validity* [36]. Social validity helps to establish whether the intervention in its current form can be utilized to improve the health situation in a given social setting, based on the viewpoint of those who will use it [37]. As a subjective measure, social validity is best assessed by those targeted for the intervention [37,38], and can be assessed before, during, or after the implementation of the intervention depending on the purpose of the assessment [39,40].

Several studies have evaluated the effectiveness of ACT and other acceptance-based interventions in low-resource settings [14,41,42]. However, such studies have not focused on the use of these approaches for promoting mental health in workplace settings[43]. While effectiveness-based studies help to generate evidence that supports the scientific validity of interventions [44], they do not guarantee the uptake of such interventions beyond the study stage, due to possible mismatches between intervention components and societal realities [44]. Thus, assessment of social desirability of the intervention may suggest modifications and adjustments that are required to increase acceptability among the targeted population, or to eliminate wastage of resources invested in evaluating interventions that are impractical to implement [45]. Social validity can be assessed using quantitative surveys [45] among targeted users, their families, groups, and influential others [36]. However, if the assessment goal is to generate feedback to enhance the appropriateness of a relatively novel intervention in a new context, qualitative interviews with knowledgeable stakeholders may generate more useful insights [45].

Consistent with this rationale, the aim of the current study is to assess the social validity of ACT-W as a workplace mental health promotion program for use with staff in Uganda. Specifically, the study explores how local mental health providers (MHPs) appraise the appropriateness of an ACT-W program's goals, and the suitability of its various training techniques for the Ugandan cultural context.

## Materials and methods

The study was guided by Wolf's social validity framework (SVF) for validating the social importance of interventions [35]. According to SVF, health interventions can be of social importance to the targeted community if they fulfill three main conditions: (i) *social significance of intervention goals*, referring to whether the behavioral goals being achieved by the intervention are what the society needs. If the treatment goals are socially valued, members of the society will voluntarily engage with the intervention as they see it as relevant to their needs [46]; (ii) *social appropriateness of the intervention procedures*, a parameter that tests whether the methods of delivery and mechanism of change used by the intervention are feasible and acceptable to the targeted society; and (iii) *social importance of effects*, which concerns the relevance of the behaviors being changed, and the effect it might bring to the intended recipients of the intervention. This third parameter questions whether the outcomes, including those anticipated or predicted, are helpful to society [39]. Exploring social validity through lens of a social validity framework ensured that the subject under study is not conflated with other relatable constructs such as acceptability or satisfaction, hence we were able to remain true to the purpose of the study. Besides, working through a framework provided guidelines that help to generate relevant and responsive data which enabled an in-depth inquiry [47]. The three constructs of SVF guided the design of an interview schedule, which we used to collect data from local mental health providers in Uganda and the analysis of data

Most studies assessing social validity often use quantitative Likert-type questionnaires to make evaluations of intervention goals, processes and effects [46]. While quantitative assessments are useful in generating evidence about the appropriateness of the intervention, they offer little insight into what needs to be done to improve the social validity of interventions [48]. In studies where stakeholder insights are key variables of interest in improving the social validity of interventions, qualitative methods offer broader scope and reach [48]. Furthermore, assessment of social validity is complex and subjective, requiring rigorous approaches such as semi-structured focus groups or in-depth interviews [35,39].

Accordingly, we used a case study qualitative approach to explore the perspectives of Ugandan MHPs regarding the social validity of the goals, processes and effect of the ACT-W to guide the adaptation of the intervention for use in Uganda. A case study approach allows in-depth exploration of issues in real-life settings, and is considered appropriate for use when exploring a professional's attitude to and experience of a new health initiative [49].

## Study procedure, setting and participants

We recruited n = 14 MHPs working in Kampala, Uganda. A sample between 9–17 is considered sufficient to reach saturation in this type of qualitative study [50]. Recruitment started on 1st June 2023 and ended 20th June 2023. Purposive sampling was used to select participants from six occupational categories: Organizational psychologists (n = 2), occupational health and safety practitioners (n = 2), human resource officers (n = 4), ACT practitioners (n = 2), practicing counsellors (n = 2), and graduate students of psychology (n = 2). The sampling approach helped to create a heterogeneous sample that offered varied views. The criteria for inclusion were: being involved in offering any form of mental health support to people at work, being at least 18 years of age, living and practising in Kampala, Uganda, and being willing to provide written informed consent. While most studies exploring social validity recommend the inclusion of users, providers and their families, we opted to work with providers in the first instance, given that the intervention is new to the Ugandan context. Research shows that subjective evaluation of social validity with experts generates important information when the goal of validation is to identify areas of modification that might improve acceptability of the intervention to the targeted context [51].

## The ACT-W intervention

ACT-W is a manualized workplace-based training program with flexible delivery format and session content, which was designed by Flaxman, McIntosh, and Oliver at City, University of London, United Kingdom. The program has been widely adopted in the UK to help improve workers' psychological health and well-being [52]. Rooted in the ACT approach, the goal of ACT-W is to improve psychological flexibility, which broadly refers to the ability to engage in personally valued patterns of behavior, even while experiencing difficult or discomforting thoughts and feelings [52]. ACT-W utilizes a simplified version of the ACT Matrix, which is presented as a "self-awareness tool", and designed to help workers notice the functions of their behavior. Specifically, ACT-W participants are invited to learn how to practice noticing behaviors that provide a sense of satisfaction that one is moving in a personally valued life direction, and also behaviors that are more about getting relief from (or which are under the influence of) difficult inner experiences. In this way, "noticing" is presented as the central psychological skill cultivated by the training.

Around the central ACT matrix practice, the training makes use of: cognitive defusion exercises, which are designed to help workers "disentangle" themselves from cognitions that exert an unhelpful influence over behavior; an emotional acceptance exercise, which involves

exploring a slightly difficult feeling at the level of physical sensations in the body; and values clarification techniques, which help workers notice the personal qualities they most want to express in their daily behavior, and identify small actions that provide a sense of satisfaction of expressing those qualities in different areas of life (e.g., work, health, and personal relationships).

The intervention is often delivered to groups in the workplace over four training sessions spread across four consecutive weeks (i.e., one session per week), with each session lasting around two hours. Although, the delivery format is considered flexible, with some training delivered and evaluated in a three-session format (with each session lasting for three hours) or in a one-day workshop [52]. The training content is summarized in Table 1 below.

Two members of the research team [KM and RK] who have experience delivering ACT-based interventions, and are Ugandan natives, delivered the ACT-W training to study participants following a one-day workshop format. The training was conducted in English and lasted for eight hours with participants taking occasional breaks to refresh and reflect on the content. After the training, three research assistants, who were trained in collecting qualitative data, engaged each participant in a one-on-one in-depth interview, which was guided by a semi-structured interview guide developed from the social validity framework S1 Text. Interviews lasted between 30 to 40 minutes and were audio-recorded.

## Ethical approval

The study received ethics clearance from Makerere University's School of Health Sciences Research and Ethics Committee (MAKSHSREC-2023-531). Although the study was considered to be of minimal risk, written informed consent was obtained. Participants were also compensated US $13 for participation. Furthermore, to protect participant's privacy, names were removed, and identification numbers were assigned during the analysis and reporting of results.

## Data management and analysis

All 14 interviews were transcribed verbatim by two Masters' students who worked as research assistants. The lead researcher [KM] sampled the transcripts and listened to the accompanying audio recordings to ensure accuracy. Participants also read through their transcripts to ensure there was no misrepresentation. Once transcripts were tested for accuracy, analysis of data began.

Data were thematically analyzed using a deductive approach following *prior* themes from the SVF. A deductive approach means applying pre-determined themes to the data. The themes can be generated from a theory, framework, literature, research problem or emerging

**Table 1. Example content of ACT-W sessions.**

| Session | Content |
|---|---|
| Session one | Introduction to mindfulness (connecting body to mind); clarifying personal values (in the form of personally valued behavioral qualities), and identifying small actions that elicit a sense of satisfaction of expressing those qualities. |
| Session two | Reconnecting body to mind, exploring the two "modes of mind" (sensing/ thinking), creating deliberate awareness on how and when to shift modes; developing skills for relating to unhelpful thoughts. |
| Session three | Reconnecting body to mind; developing noticing skills; developing skills for relating to difficult moods and emotions; viewing life from a broader perspective. |
| Session four | Mindfully making choices to break habits of autopilot; overall review and reflections of the training; setting intentions for continued noticing practice after the training. |

key assumptions of what the data may entail [53]. A deductive approach was appropriate because when the construct under study is well grounded in literature and is explicitly explained by a theory, model or framework, the analysis can then be based on the inference of the constructs of such a theory, model or framework [54]. Social validity as a construct is comprehensively explained by Wolf's SVF through the three domains (goals, processes and effect) and there is adequate literature that links the definition and conceptualization of social validity to the SVF [38,39,55]. Furthermore, several studies on social validity of behavioral interventions have used a deductive approach to data analysis following the SVF [56–58]. Further still, a deductive form of analysis is appropriate if the purpose of research is to do a detailed analysis of some aspects of the data [54,59]. Literature has also showed that social validity of behavioral change programs is best established through iterative a *priori* steps which is synonymous with the deductive approach [56]. Two authors (KM and RK) analyzed the data over six steps. In step 1, both analysts read through all 14 transcripts to familiarize themselves with the data and also to identify relevant demographic information. In step 2, the analysts selected three transcripts which they independently open-coded to generate initial codes. In step 3, the analysts reviewed and discussed the emerging codes, defined and agreed to the code definitions, and came up with a set of final codes to be used for the final analysis S1 Data. In step 4, the analysts aligned relatable codes under the three *priori* themes of the SVF (social goals, social procedures, and social effects) to create a comprehensive analysis framework, which was then applied to the remaining transcripts. In step 5, the analysts used NVivo version 12 to help analyze the remaining transcripts. The analysts remained open to including other codes that emerged as the analysis proceeded. In the final step, once the analysis was completed, writing of results commenced.

## Trustworthiness of data

To ensure trustworthiness of data, several techniques were used. First, the development of the data collection tool (interview guide) was supervised by an expert in qualitative research. Second, since the principal investigator maintained occasional contact with study participants during delivery of the ACT-W session, he did not participate in interviewing to avoid biasing respondents. Other members of the research team at master's degree level who had been trained in qualitative data collection conducted the interviews. Furthermore, the study team kept close contact with the participants and after transcribing, so that participants had a chance to review their transcripts to ensure there was no misrepresentation. Finally, the research team held several debriefs during data collection.

## Findings

**Participant characteristics.**  The age of participants ranged from 28 to 43 years with an average age of 35 years (SD = 4.75). Nine people identified as male and five as female as the sex assigned at birth. The majority had a bachelor's degree as their highest level of education (n = 8). Participant characteristics are shown in Table 2 below.

Results are reported under the three themes of the SVF as shown in Table 3 below.

**Social significance of ACT-W goals.**  When validating the social significance of ACT's goals to Ugandans, participants recognized that ACT-W aims at improving the wellness of people and applauded the importance of wellness goals. They noted that recently, employee wellness initiatives have been gaining momentum in Uganda, and many organizations desire having both physically and mentally sound employees. They further acknowledged that although organizations are starting to invest in employee wellness programs especially the physical aspect, mental wellness lags behind, making ACT-W important. Here's what some of the respondents said,

**Table 2. Social demographic characteristics of participants.**

| Demographic characteristics | n | % |
|---|---|---|
| Sex | | |
| Male | 9 | 64 |
| Female | 5 | 36 |
| Age | | |
| 26–30 years | 3 | 21 |
| 31–35 years | 3 | 21 |
| 36–40 years | 6 | 43 |
| Above 40 years | 2 | 15 |
| Highest education level | | |
| Bachelor's degree | 8 | 57 |
| Masters' degree | 6 | 43 |

*"The program is related to the wellbeing of people which is an important aspect in Ugandan workplaces which is mostly neglected"* (AW004, 30-year-old female MHP)

*"Most organizations I have worked for agree that emotional wellness of workers should be promoted but they don't know how best to do it. They say they organize physical exercises after work but they haven't done much for emotional health"* (AW009, 39-year-old male)

Participants further identified that ACT-W aims to improve the mental health of people at work, and acknowledged that although mental health used to be a neglected aspect in Uganda, the outbreak of COVID-19 and its aftermath highlighted the relevance of mental health in the country; and it's now a concern for most organizations; as stated by AW014, 36-year-old MHP,

*"Since COVID time, there has been an increased attention to mental health in the country, everyone is now speaking about mental health and workplaces are continuously holding mental health talks. Interesting how it has now become a main thing"*

Relatedly, participants were however concerned about the stigma surrounding mental health in Uganda, and anticipated that this could be a big challenge in accepting mental health

**Table 3. Adopted SVF themes and codes identified from the analysis.**

| SVF Constructs/Themes | Codes |
|---|---|
| Social significance of ACT-W goals | Importance of ACT-W goals |
| | Importance of mental health |
| | Nature of ACT-W goals<br>Relevance of self-regulation skills |
| Social appropriateness of ACT-W procedures | Acceptance of methods |
| | Language of the program |
| | Training aids and metaphors |
| | Program structure |
| | Content and concepts |
| | Believability of the treatment process |
| The social importance of ACT-W outcomes | Health enhancement |
| | Improved productivity |

promotion programs such as ACT-W. Some noted that mental health is loosely translated as having mental disorders or personal emotional problems which cannot be openly discussed, especially among men. Participants suggested a need to incorporate components of mental health awareness in ACT-W to help neutralize stigma if ACT-W is to be used in Uganda. This is exemplified by the following quotes,

> *"I see clients who often tell me how they find it difficult to talk about their struggles for fear of being judged. This is common among men who do not want to be seen as weak"* (AW014)

> *"Mental health is not well understood here, many people still relate it to being mad due to witchcraft and personal problems. A program of this nature should include awareness and fighting mental health stigma as part of the goals"* (AW002, 29-year-old-male MHP)

Participants further evaluated the nature of goals targeted by ACT-W, and observed that the intervention mainly looks at achieving *individual* goals or values. While this is considered important universally, they were skeptical that it comes at the expense of communal values, which are cherished in Uganda and many other African settings. Incorporating group values alongside individual values was recommended if the social importance of ACT-W is to be improved,

> "*Focusing solely on individual values and goals may not be well-received in a collectivist culture"* (AW005, 36-year-old male)

> *"Program is mainly focused on the individual achieving personal goals, that works in the West, in Africa we live as a collective society"* (AW002)

> *"Community activities or cooperative tasks, where individuals work together to achieve a common goal while acknowledging challenges are more appropriate and need to be included in this training"* (AW001, 33-year-old male)

Finally, some providers observed that ACT-W aims to develop self-regulatory skills which help people deal with life challenges. They recognize that in resource-constrained settings, where access to mental health services is expensive, mastery of self-management skills offers an alternative that can improve well-being of people,

> *"I realize that this program targets the development of self-management skills so that even when a person meets challenges, he /she can help him/herself survive. This approach is important for societies like ours where getting a therapist is expensive"* (AW014)

> *"The aim of the training is to equip one with skills to manage themselves even when the situation is disturbing. Many Ugandans will like this because a lot is going on and they keep it to themselves* (AW011, 32-year-old female MHP)

**Social appropriateness of ACT-W procedures.**   Under this theme of the SVF, we validated whether the methods used by the program, delivery approach, and the mechanism of change underlying ACT-W, are socially acceptable in Uganda. Participants expressed confidence that some processes used by ACT-W, such as allowing and being open to experience, are similar to religious teachings and thus are likely to be welcomed in Uganda. But participants also raised concerns that mindfulness practices, which are often used as part of ACT-W training, risk being associated with Buddhism and gauged as contrary to Christianity and Islam (the main religious doctrines in Uganda):

*"Part of the training has been about allowing uncomfortable feelings to be experienced and accepting situations as they come, this is also taught in several religions here, many people will believe it"* (AW007, a 38-year-old female)

*"Mindfulness is more of Buddhism which contradicts the strong religious links Ugandans have with Christianity and Islam"* (AW003, a 31-year-old male)

Relatedly, participants observed that, in some communities in Uganda, spirituality and traditional beliefs are still strong, and they are the main explanations for mental health-related conditions. Thus, unless ACT-W incorporates some of those beliefs, it risks conflicting with them, as stated by AW006, a 43-year-old male,

*"Some individuals and communities in Uganda have deep-rooted beliefs that prioritize supernatural explanations for mental health and are skeptical of Western therapeutic approaches"*

*"Psychotherapy in Uganda has been around long enough but not fully accepted. It's often mocked by people who think spirituality or traditional approaches offer quick solutions"* (AW010, a 28-year-old female)

*"It is important to ensure that the ACT program is respectful of religious beliefs, possibly integrating elements of spirituality where it aligns with the overall objective of the training"* (AW008, 40-year-old male)

Furthermore, while participants acknowledged that English is the official language used in workplaces in Uganda, and therefore the language in which ACT-W is written is considered appropriate for the Ugandan context, some participants observed that the ACT-W manual was written for native English speakers. Thus certain terms and jargon would be difficult to understand, which might limit uptake. Participants also emphasized that in its current form, the ACT-W language can only be understood by highly educated workers but not low-level workers who may be semi-illiterate. They recommended that trainers should conduct the training by integrating local words and phrases, and where possible simplifying the language, as indicated in the following quotes,

*"Given that English serves as the official language in Uganda and is used in most workplaces, the program will be good for high-level workers who speak English not casual workers who speak local languages"* (AW09)

*"The language of the program is a bit more advanced with difficult words, it may not be understood. If possible maybe translating to Luganda or Swahili can make it more useful to many people"* (AW014)

While participants acknowledged that it would be important to remain true to the training processes of ACT-W, they expressed concerns regarding the metaphors and other aids used in the training, considering them to be distant, foreign, and hard to comprehend. They recognize that using the existing metaphors and examples might confuse people, as indicated in the following quotes,

*"Some metaphors will be hard to figure out by people here since they are Westernized, it would be better if we use traditional Ugandan games or activities that involves cooperation and competition, highlighting the balance between accepting emotions and taking purposeful actions"* (AW009)

*"The approach is good on its own but does not incorporate traditional wisdom or proverbs that highlight the value of accepting challenges to grow stronger. It would make sense if it integrated storytelling featuring local heroes who faced adversity with courage, illustrating how"* (AW005)

Relatedly, participants appreciated the program structure and the flexibility surrounding it, stating that this will facilitate implementation. They further noted that with much flexibility involved, the program will not burden participants and trainers, and most likely they will enjoy it,

*"Flexibility of the program will make it easy to implement. There is no need to cram things, ACT feels like it is naturally flowing, and people will like it. Doesn't burden like other work training programs"* (AW011, a 32-year-old female)

*"The step-by-step breakdown of each session, coupled with detailed explanations and experiential exercises makes understanding easier"* (AW001)

Participants however expressed concern about the length of the training manual and a lot of reading which is required of trainers. They note that Ugandans may find it hard and challenging to do additional reading to appreciate the program; as stated by AW007,

*"I see this program requires a lot of reading, every time they mention extra reading materials, which Ugandan will devote all that time to reading?"* (AW007)

Relatedly, the program having many sessions (four) was also identified as a barrier to its implementation. Participants acknowledge that most workplaces in Uganda are understaffed and people work for long hours; thus, finding time for continuous sessions within a work program may prove unrealistic,

*"If this is to be implemented as a work program, and will require people to attend all these sessions on different days, I assure you employers will not accept. From the COVID time, many organizations now have few employees who work a lot to cover the gap. No space to breathe"* (AW005)

Furthermore, participants observed that ACT-W pays limited attention to power dynamics and gender within the workplace when suggesting the composition of training groups. They highlight that the program assumes a workplace to be a balanced entity where people are free to join and express themselves in a training group. They acknowledge that in Uganda, power and gender influence group interactions and must be considered, especially for programs that discuss health-related matters as typified in the following quotes,

*"Ugandan workplaces have distinct power structures that can impact how employees interact with each other. This should be taken into account"* (AW001)

*"Lower level employees will never express themselves freely in the presence of bosses"* (AW006)

*"I doubt if men will like to talk about matters of mental health in the presence of female colleagues, they won't see this as a safe space to talk about feelings"* (AW006)

Finally, regarding content and concepts used in ACT-W, participants considered the training content to have been designed for a population with good awareness of psychological

processes and mental health. They acknowledged that they were able to understand the training because of their background as trained mental health professionals, but worried about the ability of local people in Uganda with little or no awareness of mental health to understand the training:

> *"The program uses advanced psychology terms and assumes an audience with some knowledge about mental health. Most people in Uganda come to know about mental health when they develop disorders"* AW011

> *"I was able to understand the program because I am a practitioner, how about our ordinary people who think psychology is mind reading, will they understand*? *The program should be simplified"* AW003, a 31-year-old male.

**The social importance of ACT-W outcomes.** With this theme of the SVF, we looked at the relevance of the outcomes of the ACT-W training to Ugandans. Since the ACT-W program aims at promoting psychological flexibility to influence well-being and mental health, participants commented on the importance of such outcomes to society in Uganda.

Participants noted that living in a resource-constrained setting presents many life challenges that impact people's wellbeing. Thus, participants noted that an important outcome of this program is that it promotes wellbeing by helping people develop the self-management skills needed to cope with such challenges:

> *"Living in a poor country is challenging, we struggle with so many things and a lot is going on in people's lives. A program that can help people experience a better life even with their problems is so important"* (AW004)

> *"Of late we see many people breaking down due to stress over survival, work, relationships, disease and many things. Anything that can support people to live better is relevant"* (AW0011)

As a workplace program, participants also indicated that when people experience this program and live a better life, their personal and work productivity may improve, which in turn helps their organizations,

> *"We all know that mental wellness affects the productivity of individuals, this program is training people to handle their mental health. I am sure they will turn out to be more productive for themselves and their organizations"* (AW004)

## Discussion

The purpose of this study was to assess the social validity of ACT-W as a workplace mental health promotion program for potential use in Uganda. Specifically, the study sought to explore how local mental health providers appraised the appropriateness of this workplace program for the Ugandan context, to ascertain fit before potential wider workplace implementation. The social validity framework was used to ascertain the likely acceptability of the intervention's goals, the congruence of program effects with the needs of this society, and the acceptability of the training procedures and techniques to the Ugandan population. These findings are intended to guide adaptation of ACT-W for contextual relevance and cultural salience in preparation for implementation within Uganda.

The findings indicated that Ugandan MHPs had confidence that this type of workplace intervention would be a useful addition to current efforts to support mental health. They acknowledged promoting wellbeing and mental health as important goals to Ugandans, applauded the development of self-management skills as necessary due to the challenges associated with accessing professional services, and considered the flexibility involved in the delivery of ACT-W as key in easing implementation. However, concerns were raised about the intervention promoting individuals' goals in a communal society, use of meditation practices (which are linked to Buddhism), and use of unfamiliar metaphors, highlighting a need for adaptations to some of the training content, methods, and language. To our knowledge, this is the first study to gauge the social appropriateness of this type of acceptance-based program as an alternative workplace mental health promotion initiative for use in Uganda. Although related psychotherapies have previously been evaluated in Uganda, such programs have not been targeted or tailored towards the workplace setting, and have not been subjected to social validation by professionals who operate in that specific context. Overall, the goals, processes and outcomes of ACT-W were perceived to be socially valid. Nonetheless, this research also revealed specific aspects of the training techniques and procedures that would need to be modified to ensure the ecological validity of the ACT-W approach within the Ugandan workforce.

When socially validating the goals of ACT-W, results showed that the promotion of mental health and emotional wellness were of great relevance to Ugandans, and have recently gained increased attention in various workplaces. The precarious work environment characterized by longer working hours, work overload, low pay and poor working conditions coupled with high rates of poverty and the difficult living conditions in the country have increased mental vulnerability among the population. Accordingly, interventions targeting mental health are viewed as socially relevant. The study's findings are supported by literature from low-resource settings, which identifies mental ill-health to be a public crisis, and suggests that mental health support programs are relevant for low-resource environments [7]. The relevance of mental health care as an important goal for Ugandans is further emphasized by a recent study that showed that approximately 14 million Ugandans have mental health-related challenges; while access to mental health care remains a significant challenge due to resource constraints, necessitating consideration of alternative cost-effective programs [60].

Although the goals of ACT-W were well received, the intervention's focus on individual values rather than collective values was perceived to be inconsistent with the needs of a collective society like Uganda. In this more socialistic setting, people evaluate and validate their values and practices based on the impact on others, and such practices are reflected in the relationships that people form. Thus, any successful psychological program to be used in a collective society should be mindful of the importance of collective values. This finding concurs with literature that highlights how Western psychotherapies often have a bias towards individualism, which complicates their application in non-Western cultures [61].

In addition, the participants' reflections are consistent with studies indicating that when mindfulness is applied in ways that solely focus on individual benefits rather than communal benefits, it creates tension because traditionally mindfulness was designed to be a community practice prioritizing collective values [62]. Similarly, other studies acknowledge that while expressing individual values in one's action is at the heart of many ACT-based interventions, such personally relevant goals are affected by socially normative pressures, especially in socially cohesive societies, underscoring a need to balance individual achievements and collective goals [63]. Thus, there appears to be practical utility in incorporating collective values in ACT-W in this cultural context. One advantage of the ACT approach is that it has been appraised to be flexible and open to continuous adaptation to respond to the needs of various communities [64].

As for specific training procedures and practices, while the methods used in ACT-W training were considered to be socially acceptable in Uganda, several adjustments need to be made if the contextual needs of Ugandans are to be fully catered for. Examples of needed adjustments include: using mindfulness exercises that do not seem to contradict religious beliefs; relating the training examples and scenarios to religious and spiritual practices; simplifying the language of the training manual (to be used by local providers); making direct translations of the content to the most popular local dialects; developing locally understood metaphors with local aids or content that is comprehensible and relatable; and including a session that promotes mental health awareness in the early phase of the training. Such adaptations are broadly supported by research denoting that integrative therapies that incorporate psychological, spiritual and diverse cultural perspectives advance the evolution of interventions to ensure any benefits are spread across more diverse populations [65,66].

These findings are further supported by studies that emphasize the need to customize delivery methods and procedures to societal preferences, if contextual behavioral science initiatives are to be accepted in varied societies [67]. Participants applauded the flexibility afforded by the ACT-W approach, particularly for allowing workplace trainers to cultivate ACT's processes without a rigid guideline to follow a specific pattern. Similar recommendations can be found in the literature, emphasizing that processes and procedures used in ACT-based interventions need operational adjustments to fit requirements of the subject and context under consideration if they are to become socially valid for such settings [55].

Another key finding of the current study is that local MHPs considered ACT-W to be a program developed for audiences with high mental health awareness. The content and processes of the program assume a knowledgeable participant which poses a limitation when used in a society like Uganda where mental health awareness is very limited. The lack of awareness surrounding mental health affects help-seeking behaviors and acceptance of support interventions. Thus, programs that target mental health and emotional well-being may need to start with promoting mental health awareness. This finding aligns with several studies that recognize a lack of mental health awareness among Ugandans, and with calls for the promotion of mental health awareness [60, 66].

Finally, ACT-W was viewed by these mental health providers as advancing socially important effects for both individuals and organizations, particularly because of the perceived utility for inducing behavior change that could drive general well-being and productivity improvements. More broadly, the results indicate the relevance of this type of alternative worksite wellness strategies to complement existing approaches to improving adult mental health in Uganda. This was an encouraging theme observed across this study's data, given that uptake is likely to be enhanced when outcomes of an intervention are perceived to be desirable [68].

## Strengths, limitations, and future directions

A key strength of this study stems from the close collaboration with a sample of local professionals, who clearly possessed knowledge and expertise around promotion of mental health in Ugandan workplace settings. The study's specific findings and recommendations demonstrate the value of performing a cultural translation of this type of psychological intervention, prior to advocating widespread implementation in this particular context. Given the difficulties in developing home-based interventions for every different context, gauging the appropriateness of an already developed and evidence-based intervention approach to a new context may serve to increase global access to this type of program, while also saving considerable time and resources which would otherwise need to be invested in developing new programs. This strategy of identifying aspects that need modification to make ACT-W suitable for use in Uganda

could ultimately lead to the development of a tailored, implementable, and popular workplace-based mental health promotion program.

Relatedly, the study relied on stakeholder engagements to assess the ecological validity of ACT-W. This approach ensured that knowledgeable MHPs who are natives of the community gave expert feedback to gauge the social appropriateness of ACT-W in a Ugandan setting. This approach ensured that those who are likely to deliver the intervention were validating it. Thus, the process was done with them, not for them. Relying on professionals within the community meant that we were able to obtain informed feedback that integrated practice and context. As ACT-W is a new intervention in Uganda, this initial assessment from people with a deep understanding of psychological processes was deemed necessary. The program tailoring recommendations gathered from this sample of local mental health practitioners seem likely to improve the "fit" of the ACT-W program with the cultural context, thereby increasing the likelihood of effective implementation.

Our study also has some limitations. First, the MHPs participating in this study were all recruited from one region (Kampala). This means that their voices do not necessarily represent the various regions in Uganda. Uganda is culturally diverse, and urban settings might have different contextual requirements from rural settings. Hence, we cannot yet be sure that that the findings are similarly relevant and representative of the various contexts in Uganda. The natural diversity within Kampala as the capital city arguably supports the validity of these professional's opinions. We suggest our findings could be considered as pilot results, which could now inform the next stage of adaptation, testing, and implementation of ACT-W within Ugandan workplaces. Given that adaptation is considered an iterative and continuous process, further modifications to suit specific Ugandan communities can be incorporated as this process progresses. One advantage of focusing on the social validity of ACT-W, is that this program was itself explicitly designed to be technically and structurally flexible. As a result, the program has continued to be adapted to different aims and occupational contexts in high-income countries, and continues to integrate practice innovations in the ACT approach [19].

A second limitation is that the study relied exclusively on feedback from mental health providers. While this approach ensured we gathered "expert" feedback, it did not involve potential end-users of the focal workplace training program (i.e., Ugandan workers who might be interested in receiving this type of training). Professional evaluators might be biased by their educational backgrounds, and look at mental health issues differently from community potential end-users. Nonetheless, it is noteworthy that professional judgment is often preferred during the initial stages of an intervention's adaptation [51].

We believe this study's findings highlight three immediate and useful modifications to the ACT-W program. First, it seems relatively straightforward to identify and include more examples of collective values in this type of worksite training (e.g., self-sacrifice, being dependable, helping others), while reducing examples that may be more personally meaningful in individualistic cultural contexts (e.g., independence, autonomy, self-reliance). Second, the centrality of the ACT matrix in this program, means that any "formal" mindfulness meditation practices could simply be omitted, on the grounds that the non-judgmental and curious "noticing" that is practiced throughout this program would (in theory) enhance mindfulness processes [69,70]. Third, metaphors and images used in the training can be substituted with more familiar cultural references and examples.

## Conclusion

In numerous high-income countries, workplace training programs derived from acceptance and commitment therapy (ACT) are being delivered and endorsed to help improve people's

mental health [71]. Workers in LMICs rarely have access to this same type of evidence-based mental health promotion program, in part because of the lack of culturally sensitive adaptations. The current study demonstrates the utility of engaging local experts in gauging social appropriateness of this workplace intervention approach for a new setting. The findings revealed that, while ACT-W was considered to be socially relevant for Uganda, careful adaptations to intervention content and methods need to be applied to enhance this program's social validity. Given the promising and useful findings generated by this research, our aim is to progress to the next stage of co-creating and empirically testing a culturally adapted ACT-W program for use with staff in Ugandan organizations.

## Supporting information

**S1 Text. Interview guide.**
(PDF)

**S1 Data. Study data.**
(DOCX)

## Acknowledgments

The authors wish to thank Isaac Mukwaya and Dennis Lisbon Kiore for supporting data collection and all the mental health experts that participated in the study.

## Author Contributions

**Conceptualization:** Khamisi Musanje, Paul E. Flaxman, Rosco Kasujja.

**Data curation:** Ross McIntosh.

**Formal analysis:** Khamisi Musanje, Rosco Kasujja.

**Funding acquisition:** Paul E. Flaxman.

**Investigation:** Khamisi Musanje, Paul E. Flaxman, Ross McIntosh, Rosco Kasujja.

**Methodology:** Khamisi Musanje, Rosco Kasujja.

**Project administration:** Khamisi Musanje.

**Resources:** Ross McIntosh.

**Supervision:** Paul E. Flaxman, Rosco Kasujja.

**Writing – original draft:** Khamisi Musanje, Rosco Kasujja.

**Writing – review & editing:** Khamisi Musanje, Paul E. Flaxman, Ross McIntosh.

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
