## [Decision Letter · Decision Letter 0]

2 May 2024

PMEN-D-24-00104

Social validity of acceptance-based workplace mental health training for use in a low resource setting. A qualitative study with Ugandan mental health providers

PLOS Mental Health

Dear Dr. Musanje,

Thank you for submitting your manuscript to PLOS Mental Health. After careful consideration, we feel that it has merit but does not fully meet PLOS Mental Health’s publication criteria as it currently stands. Therefore, we invite you to submit a revised version of the manuscript that addresses the points raised during the review process.

We look forward to receiving your revised manuscript.

Kind regards,

Nancy Clark, PhD

Academic Editor

PLOS Mental Health

Journal Requirements:

Additional Editor Comments (if provided):

Reviewers' comments:

Reviewer's Responses to Questions

**Comments to the Author**

1. Does this manuscript meet PLOS Mental Health’s publication criteria? Is the manuscript technically sound, and do the data support the conclusions? The manuscript must describe methodologically and ethically rigorous research with conclusions that are appropriately drawn based on the data presented.

Reviewer #1: Yes

2. Has the statistical analysis been performed appropriately and rigorously?

Reviewer #1: N/A

3. Have the authors made all data underlying the findings in their manuscript fully available (please refer to the Data Availability Statement at the start of the manuscript PDF file)?

Reviewer #1: Yes

4. Is the manuscript presented in an intelligible fashion and written in standard English?

Reviewer #1: No

5. Review Comments to the Author

Reviewer #1: Thank you for submitting your work to Pmen Jr.

The data was analyzed by the writers using a deductive thematic analysis approach. Nevertheless, no particular literature or theory is offered in the method section to clarify the meaning of a deductive thematic approach or the reasons it might be preferred to other approaches. This lack of backing calls into question the rationale and preferences for this kind of approach.

A deductive thematic analysis is a methodical and structured procedure that entails finding and classifying themes or patterns in the data. The data must then be fully explained to readers with references to pertinent literature. The analysis process is guided by an established body of information or predetermined theoretical framework at the outset. For instance, figures, tables, and charts

As I mentioned before, the approach description is included in the theme analysis together with the literature review, but this is not enough support. Before producing an extensive literature review with citations from the most recent to the earlier studies, authors must first distinguish the introduction from the literature review.

Authors must first distinguish the introduction from the literature review in order to improve the support offered by these two techniques. The introduction gives a broad background for the research as well as a succinct summary of the subject. It does not, however, particularly address the goals or research questions that will direct the literature review. Writing the introduction and literature review separately allows authors to make sure each piece does what it is supposed to do.

Writers can concentrate on producing an in-depth literature evaluation after separating the introduction from the review. A comprehensive literature review entails a detailed analysis of the body of existing research on the subject of interest. To give a thorough grasp of the study field, it entails finding pertinent studies, evaluating their major conclusions, and synthesizing this data.

Authors should cite a variety of sources, both historical and contemporary, to guarantee a comprehensive literature assessment. Authors can provide the groundwork for future study and expand on what is already known in the field by utilizing citations from earlier studies. On the other hand, authors can remain current with the most recent advancements and theories in the field by incorporating citations from recent research. Writing from both historical and modern viewpoints allows authors to present a complete picture of the research issue.

Authors should also assess the literature they include in the review critically. This entails evaluating the research's quality, taking into account its methodology, and determining the accuracy and dependability of the data it offers. Authors can make sure that the review offers trustworthy support for the research being done by thoroughly analyzing the literature.

6. PLOS authors have the option to publish the peer review history of their article (what does this mean?). If published, this will include your full peer review and any attached files.

**Do you want your identity to be public for this peer review?** For information about this choice, including consent withdrawal, please see our Privacy Policy.

Reviewer #1: No

---

## [Editor Report · Decision Letter 1]

13 Aug 2024

PMEN-D-24-00104R1

Social validity of acceptance-based workplace mental health training for use in a low resource setting. A qualitative study with Ugandan mental health providers

PLOS Mental Health

Dear Dr. Khamisi Musanje

Thank you for submitting your manuscript to PLOS Mental Health. After careful consideration, we feel that it has merit but does not fully meet PLOS Mental Health’s publication criteria as it currently stands. Therefore, we invite you to submit a revised version of the manuscript that addresses the points raised during the review process.

Thank you for addressing the revisions from reviewer one. There are still minor revisions to be made. Please address the secion on line 45 re deductive analysis. It is not clear how the analysis was deductive, why not inductive given the qualitative participatory nature of the interviews with the workers?

Please clarify line 205 where you discuss "lense of a framework" do you mean social validity framework as in line 516?

thank you for your attention to these details,

Please ensure that your decision is justified on PLOS Mental Health’s publication criteria and not, for example, on novelty or perceived impact.

Please submit your revised manuscript by . If you will need more time than this to complete your revisions, please reply to this message or contact the journal office at mentalhealth@plos.org. Please include the following items when submitting your revised manuscript:

We look forward to receiving your revised manuscript.

Kind regards,

Nancy Clark, PhD

Academic Editor

PLOS Mental Health
---

## [Editor Report · Decision Letter 2]

20 Aug 2024

Social validity of acceptance-based workplace mental health training for use in a low resource setting. A qualitative study with Ugandan mental health providers

PMEN-D-24-00104R2

Dear Assistant Lecturer Musanje,

We are pleased to inform you that your manuscript 'Social validity of acceptance-based workplace mental health training for use in a low resource setting. A qualitative study with Ugandan mental health providers' has been provisionally accepted for publication in PLOS Mental Health.

Best regards,

Nancy Clark, PhD

Academic Editor

PLOS Mental Health